# A new magnetic observatory in La Réunion Island – meeting data quality requirements in a volcanic island setting

Benoit Heumez[1], Frédérick Pesquiera[2], Abdelkader Telali[1], and Vincent Lesur[1]

[1]Université Paris Cité, IPGP, CNRS, France
[2]Université Paris Cité, IPGP, CNRS, France

**Correspondence:** Lesur (lesur@ipgp.fr)

**Abstract.** A new magnetic observatory has been set on La Réunion island in the Indian Ocean through a collaboration between the "Institut de physique du globe de Paris" (IPGP) local volcano observatory (Observatoire Volcanologique du Piton de la Fournaise – OVPF) and its magnetic observatory service. This observatory is isolated and serves for monitoring the evolution of the Earth's magnetic field in that region. It is also particularly useful for large scale modelling of the core field and other contributions to the geomagnetic field. Three-component vector magnetic field data are continuously collected at 1Hz using a fluxgate, while scalar data are collected at 0.2Hz with a proton magnetometer. The data are transmitted every 5 minutes to IPGP main site and made immediately available to the scientific community (see `www.bcmt.fr`). Due to the strong magnetic field generated by the surrounding volcanic rocks, the differences between the magnetic field strengths as recorded by the proton magnetometer and the strengths calculated from the recorded vector field values vary by more than $\sim$2 nT during a day. To circumvent this difficulty, constant offset values of -2400 nT, 280 nT and -20 nT are added to the $X$, $Y$ and $Z$ magnetic field components respectively, prior to the data distribution. We show that this approach efficiently reduces the differences between measured and calculated magnetic field strengths inside a day. Calibrated observatory data have been calculated over the year 2023 and, although the baseline values present variations up to 70 nT throughout that year, the derived data meet the quality required for an INTERMAGNET observatory. A Fourier analysis of the data shows that these are not contaminated by a significant noise even if peaks at 0.2Hz indicate a small cross-talk between vector and scalar instruments.

## 1   Introduction

There are currently around 120 magnetic observatories around the world collecting data, most of them being part of INTER-MAGNET (Love and Chulliat, 2013); an international organization promoting high quality standards for magnetic data and acquisition processes, and free data distribution (`www.intermagnet.org`). These observatories allow to monitor the geomagnetic field changes over decades and serve not only for the study of the core magnetic field but also the fields generated in the ionosphere and magnetosphere together with those generated by their induced counterpart currents in the conductive bodies inside the Earth. Numerous other natural sources contribute to the observed magnetic data such as oceanic tides and currents. However, for having a global view of the magnetic field evolution, it is preferable to have a homogeneous distribution of observatories over the world (e.g. Langel et al., 1995). This is far from being the case with most of observatories being

located in Europe and Northern America while only few observatories are set in the Middle East regions, Africa and South America. Despite very difficult climatic conditions there are several observatories in Antarctica in contrast with oceanic areas where only a few observatories are set on remote islands.

In the Indian Ocean, as in the Atlantic or Pacific Oceans, there are currently very few observatories (See Fig. 1). To the East, Gingin (GNG), Learmonth (LRM) and Cocos Island (CKI) INTERMAGNET observatories, all under Australian institution responsibilities, are producing data. To the North, India is running several INTERMAGNET observatories: Alibag (ABG), Hyderabad (HYB), Choutuppal (CPL). There is also the Gan International Airport observatory (GAN) in Maldives Islands that has been set in 2012. The observatory in Antananarivo (TAN), stopped producing calibrated data in December 2007. A new observatory has been set in Fihaonana (Madagascar) but is not yet distributing data. Further to the West data are distributed by the Maputo (LMM) and Nampula (NMP) observatories in Mozambique up to 2017 and 2019 respectively, and by Hartebeesthoek (HBK) observatory. To the South, three observatories on the Kerguelen, Crozet and Amsterdam islands have not been delivering calibrated data to Edinburgh World Data Center (wdc.bgs.ac.uk) since 2013, 2015 and 2013, respectively, although some variation – i.e. non-calibrated, data are available on the BCMT data repository (www.bcmt.fr). To enhance the spatial coverage of observatories in the central Indian Ocean, a new observatory was established on La Réunion Island, approximately 875 km to the east of the former Antananarivo observatory and more than 1800 km away from the closest currently active observatory in Nampula (Mozambique). The choice of this island comes primarily from the predicted evolution of main magnetic field given by the International Geomagnetic Reference Field, version 13 (IGRF-13), that forecasts a maximum increase of the south hemisphere magnetic field strength in this area. Besides, the "Institut de physique du globe de Paris" (IPGP) already runs a volcanic observatory in La Réunion to monitor the Piton de la Fournaise volcano. The scientists and technicians of the volcanic observatory eased the installation of the magnetic observatory and furthermore provide the required scientific and technical expertise to perform the weekly handmade absolute measurements. The presence of the magnetic observatory on this island is also a new asset for processing magnetic survey or variometer station data acquired for monitoring the volcano activity.

Building an observatory on an isolated island usually comes with specific challenges as these islands are typically of volcanic origins and therefore are made of rocks presenting strong magnetization. It follows that, contrary to the traditional continental setup, observatories on these islands are often set in areas of strong magnetic field gradients. Such gradients do not preclude accurate measurements of the magnetic field strength and direction when modern instruments are used, but it is nonetheless difficult to reconcile the data acquired on different locations of the observatory site. These data are continuous series of vector magnetic field measurements made using fluxgate magnetometers, series of total field strength generally obtained with a proton or optically pumped absolute magnetometer, and handmade absolute measurements typically made on a weekly basis. These three types of data are made at different places, few meters apart, and have to be processed to give a continuous series of calibrated vector magnetic data located on the absolute reference pillar of the observatory. It is this continuous series of second or minute mean calibrated data that is ultimately distributed by the observatories for scientific or technical applications.

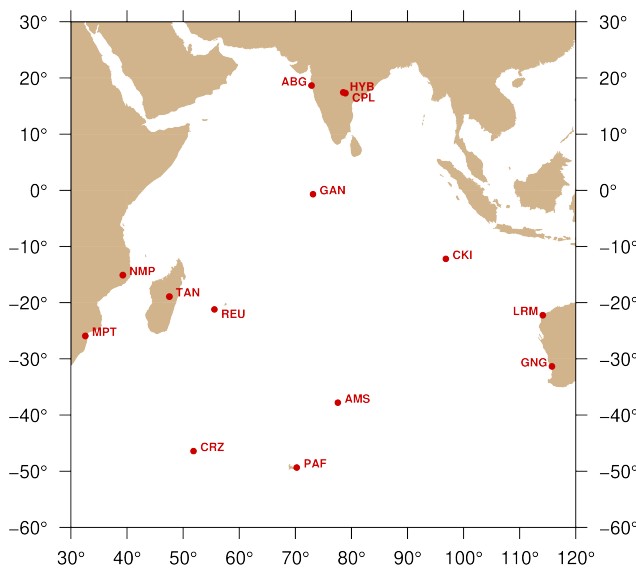

**Figure 1.** Map of magnetic observatories in the Indian Ocean that have released data in recent years.

In the next section the new observatory location and setting are described in detail. In section 3 the data processing applied on the vector data to estimate the field on the observatory reference pillar is presented. In the following section are shown results of one full year of data acquisition. Calibrated data for year 2023 are presented and analyzed. The last section is dedicated to the conclusion.

## 2 La Réunion observatory setting

La Réunion Island, located in the Indian Ocean, is a volcanic island characterized by strong magnetic anomalies and steep gradients due to highly magnetized rocks. Several sites were considered for setting the magnetic observatory in the ?Plaine des Caffres?, a smooth and relatively flat area, south of the island, between the two volcanoes. Aeromagnetic surveys conducted over the area indicated relatively low gradients in the area. To refine site selection, we carried out both vertical and horizontal gradient surveys at candidate locations. The resulting grid maps and spot measurements of the total magnetic field revealed the heterogeneous nature of the volcanic magnetic environment, highlighting the challenges inherent in establishing an observatory in such a setting. The observatory installation was completed within one year in three main stages : surveys of potential sites, pillars construction and equipment deployment. The chosen site at 21°12'21.2" S, 55°34'35.3" E and 1580 m altitude, is as isolated as possible and situated at 500 m from the volcanic observatory, where observers trained for absolute measurements and basic maintenance are available. The land, covered by forest, is owned by the national forestry office. A 9-year agreement has been signed between our institutes. The installation needed to be as little intrusive as possible but autonomous. The area has little elevation change, the forest is not dense and no endemic trees are present allowing us to clear the surrounding vegetation

to guarantee the sun exposition of solar panels. A visual target was installed on a single concrete pole for geographical reference at a distance of about 40 m from the observatory main pillar. As secondary target, a natural peak on the volcano at around 5 km away is used. A good grounding to avoid lightning strikes and a solidly built infrastructure to provide good resistance to hurricanes have been necessary. The constructions are free from ferromagnetic (or magnetic) materials. Materials were tested using a magnetometer before use or installation. Fibered reinforce concrete was used in place of the usual iron-reinforced concrete.

Three types of data are collected on the observatory site:

- *Variation vector magnetometer data:*
  The vector magnetic field is sampled at 1 Hz, using a DVM-19 full range three-axial fluxgate instrument built in the Chambon-la-forêt French national observatory. This instrument has a relatively low noise level ($< 15\text{pT}/\sqrt{Hz}$) and can be rigorously calibrated thanks to its full-range ($\pm 70~\mu\text{T}$) capabilities. It has, as all fluxgate instruments, a dependence on temperature that is of the order of 300 pT/$^o$C. The instrument is seated on the "variometer" pillar; because of possible movements of the pillar or slight temperature variations, the collected data cannot be seen as absolute data. The data collected form the variation vector data.

- *Variation scalar magnetometer data:*
  The scalar instrument is a Geomag SM90R, Overhauser-type, scalar absolute magnetometer sampling the magnetic field at 0.2Hz. These types of instruments are sensitive to magnetic field gradients; therefore, the scalar magnetometer was installed at a height of 1.7m above the ground and positioned several meters away from the vector magnetometer. This setup minimizes potential interference between the two instruments. These types of data are called the variation scalar data.

- *Handmade absolute measurements:*
  The handmade absolute data are collected on the observatory main pillar using a Bartington Mag01H single-axis fluxgate magnetometer probe mounted on a Zeiss 010A non-magnetic theodolite. Each absolute observation is a combination of a series of eight handmade Declination and Inclination measurements. These angle measurements are completed with absolute measurements of the magnetic field strength made using a Gemsystem GSM-90T. The technique used has been described for example in Newitt et al. (1996). These types of data are collected at least once a week.

As partly described above, two pillars in fibered concrete were build to set the instruments. A large and deep one, at chest height – i.e. 1.4 m, for absolute measurements and, $\sim 10$ m away, a small one to rest the variation vector magnetometer. This latter variometer pillar is $\sim 40$ cm above ground and is covered by a well insulated box, filled with water bottles for increased temperature stability. Such a short pillar means increased measurements stability over long time periods. However, it also means larger contributions from magnetized surrounding rocks. Our choice of a short pillar is due to the risk associated with recurrent hurricanes in this region and therefore to the requirement of a robust installation. This study demonstrates the

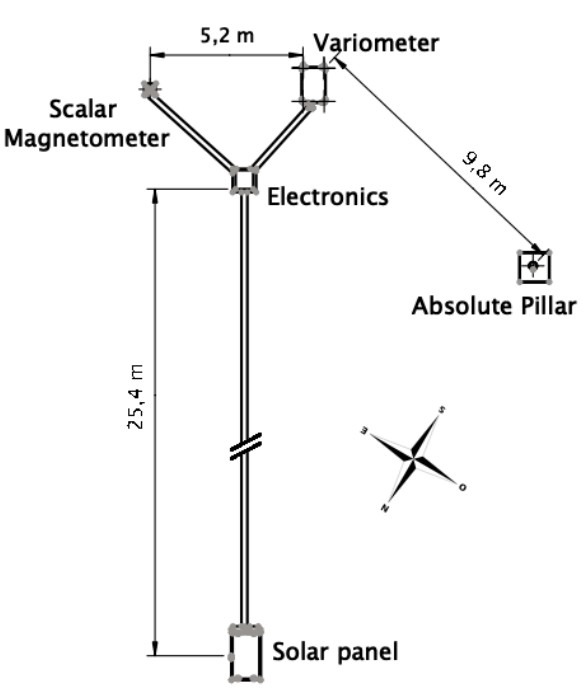 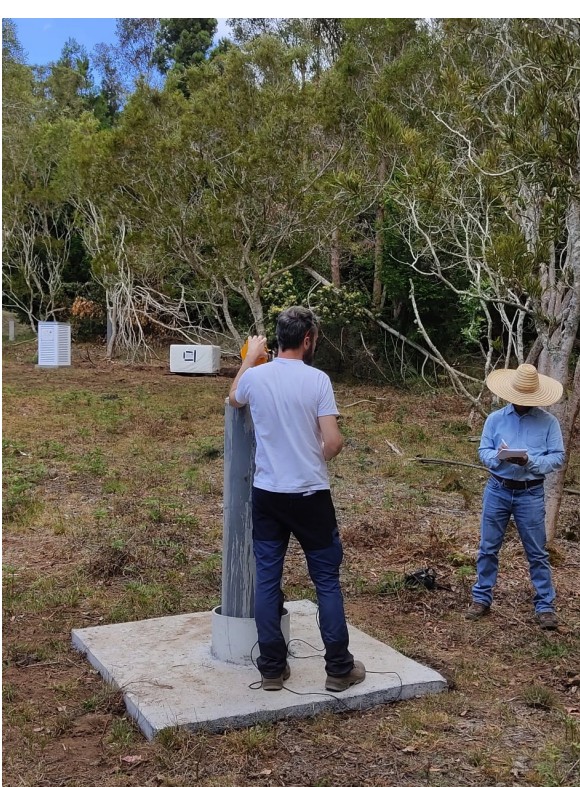

**Figure 2.** Left: a schematic map of the observatory. Right: Photograph showing the layout of the La Réunion Observatory. In the foreground, the main observatory pillar is seen during a training session. In the background, from left to right, are: the grey concrete target pole; the white meteorological box housing the sensor electronics; the grey vertical PVC tube, partially obscured by trees, containing the variation scalar magnetometer; and the plastic box containing the variation vector magnetometer, covered with a white thermal blanket.

110    methods employed to mitigate the influence of strong magnetic contributions from surrounding rock formations. The variation scalar magnetometer sits on a 1.7 m tall mast, fixed in concrete and covered in a PVC tube. This variometer pillar, the variation scalar magnetometer site and their respective electronics are roughly 6 m from each other, forming a triangle shape (See plan in Figure 2). Vector and scalar data are acquired by an IPGP in-house built "ENO4" data logger, which is based on a beaglebone platform, and transmitted to the Paris main servers via GSM digital cellular signal, typically every 5 minutes. A `mqtt` real-time
115    transmission protocol is integrated and used for monitoring purpose. The observatory has been designed to require low electric power and to be autonomous in power and communication with a single solar panel and a GSM transmitter placed 25m away from the sensors.

Variation data and absolute measurements started in December 2022. The island is subject to seasonal hurricanes/tropical cyclones, but the observatory did not suffer from the hurricane Belal in January, 15, 2024. Only the GSM connection was down for a few days. There are no frequent thunder strikes in the area. We do not foresee major difficulties in the operation of this observatory regarding its general infrastructure over the coming decade.

## 3  Data processing

### 3.1  Variation data and processing technique

The calibrated data distributed by the observatory are series of 1Hz variation vector magnetic data and 0.2Hz variation scalar data but estimated on the observatory main pillar (or main pillar) site such that they fit the handmade measurements. The differences between the magnetic field strengths computed from the variation vector measurements and variation scalar data, as estimated on the main pillar, define a data quality criteria. These differences are expected to stay within $\sim 1$ nT around zero to meet the INTERMAGNET quality standard (INTERMAGNET, 2020).

In the specific case of observatories installed in an area of strong magnetic gradient this criteria is particularly difficult to meet, even for time series of less than a day for which the pillars can be assumed to be steady and temperature variations to be negligible. Let us assume that the magnetic field at the variometer pillar is:

$$\mathbf{b}_v = \begin{bmatrix} x_v \\ y_v \\ z_v \end{bmatrix}, \tag{1}$$

where $x_v$, $y_v$ and $z_v$ are the three magnetic field orthogonal components in the magnetometer reference frame – i.e. the frame defined by the three orthogonal axis of the fluxgate magnetometer. The associated magnetic field strength is:

$$F_v = \sqrt{x_v^2 + y_v^2 + z_v^2}, \tag{2}$$

but the strength of the field on the observatory main pillar is:

$$F_p = \sqrt{(x_v + \delta x)^2 + (y_v + \delta y)^2 + (z_v + \delta z)^2}, \tag{3}$$

where $\delta x$, $\delta y$ and $\delta z$ are the differences in the magnetic vector field on the main pillar relative to the field recorded on the variometer pillar, in the same reference frame. It can be assumed that these differences are constant in time, which is consistent with the assumption that the magnetic field gradient observed on site is exclusively due to the magnetization of local rocks and is not linked to external, induced or core field signals. It can be shown (see appendix A) that, to the first order in perturbations, the magnetic field strength difference between the recorded magnetic field at the variometer pillar and its value on the observatory

main pillar is:

$$\Delta F = F_p - F_v \simeq \hat{\mathbf{b}}_v \cdot \begin{bmatrix} \delta x \\ \delta y \\ \delta z \end{bmatrix}, \tag{4}$$

where $\hat{\mathbf{b}}_v = \mathbf{b}_v / F_v$ is the unit vector giving the magnetic field direction at the variometer pillar site. As the magnetic field on the variometer pillar changes in direction over time – e.g. due to the $S_q$ current system in the dayside ionosphere, it is clear that $\Delta F$ changes with time even if $\delta x$, $\delta y$ and $\delta z$ are constant. The variations on $\Delta F$ over a day are generally small, but cannot be neglected for observatories set in areas of strong magnetic field gradients – i.e. where the $\delta x$, $\delta y$ and $\delta z$ values can reach a thousand of nanoTesla (nT) or more. This is the case for La Réunion observatory where the variometer pillar is small and therefore where the surrounding rock magnetization contributes significantly to the recorded magnetic field components (see section 2). In the longer term, for example a year, $\Delta F$ may change significantly with temporal variations of $\delta x$, $\delta y$ and $\delta z$ due to, e.g., temperature or environmental changes.

Keeping the daily $\Delta F$ variations within small values is a prerequisite for deriving definitive calibrated data at La Réunion observatory. A simple but efficient way to solve this problem is to set up a method for estimating the $\delta x$, $\delta y$ and $\delta z$ values in the sensor reference frame. This is precisely what the baseline estimation method presented in Lesur et al. (2017) does. This approach is briefly revisited in the remainder of this section.

The calibrated magnetic vector field $\mathbf{b}_p$ estimated on the observatory main pillar in a local geodetic reference frame is:

$$\mathbf{b}_p = \mathbf{R}_\theta \begin{bmatrix} x_v + \delta x \\ y_v + \delta y \\ z_v + \delta z \end{bmatrix}, \tag{5}$$

where $\mathbf{R}_\theta$ is a rotation matrix for an angle $\theta$, positive anti-clockwise, around the local vertical axis. There are four calibration parameters to be estimated, namely $\theta$, $\delta x$, $\delta y$ and $\delta z$. We assume that $\theta$ takes a constant value over several months whereas the other parameters are taking constant values only over a single day. The vertical direction on the variometer pillar is assumed to be the same as on the main pillar. It follows that $\delta z$ is independent from the $\theta$ angle, whereas the $\delta x$ and $\delta y$ depend strongly on this angle value. It is noted that the vector magnetic instrument is oriented such that its $x$-component is approximately aligned with the direction of the local magnetic North, and it follows that the $\theta$ angle is close to $(-1\times)$ the Declination angle[1] when the $\delta x$ and $\delta y$ parameters are small. However, that angle may be significantly different from the Declination when $\delta x$ and $\delta y$ parameters are large – i.e. when there are local strong gradients of the magnetic field on the observatory site.

To estimate the calibration parameter values, we require the calibrated magnetic vector field to match the absolute observations obtained on the observatory's main pillar. However, this constraint alone is insufficient for a robust estimation of the angle $\theta$.

---

[1] the Declination is positive clock-wise

Therefore we additionally require the calibrated magnetic vector field to fit the hourly spot values of the magnetic field strength $F_s$ given by the variation scalar data. This imposes to introduce a further parameter $\delta F = F_p - F_s$, also constant over one day, that describes the magnetic field strength difference between the main pillar position and the scalar variometer position. Assuming $\delta F$ constant is valid only if the magnetic field gradients are small between these two positions. This is a reasonable approximation because the main pillar and the scalar proton magnetometer are at $1.5$m and $1.7$m over ground, respectively,

and therefore are in an area of smaller magnetic gradients. This approximation was validated by placing an additional scalar magnetometer on the main pillar for one day (23rd June, 2023) to record at $0.2$ Hz the magnetic field strength simultaneously on the two locations, main pillar and variation scalar magnetometer pillar. The difference between the measurements did not exceed 0.5 nT.

Let assume that we have a set of absolute data on the main pillar, and that a rotation angle $\theta$ is chosen, then, there are two types of equations that can be used to find the daily values of $\delta x$, $\delta y$, $\delta z$ and $\delta F$:

$$
\begin{bmatrix} x_a \\ y_a \\ z_a \end{bmatrix} - \begin{bmatrix} \tilde{x}_v \\ \tilde{y}_v \\ \tilde{z}_v \end{bmatrix} = \mathbf{R}_\theta \begin{bmatrix} \delta x \\ \delta y \\ \delta z \end{bmatrix} + \begin{bmatrix} \epsilon_x \\ \epsilon_y \\ \epsilon_z \end{bmatrix}, \qquad \text{where :} \quad \begin{bmatrix} \tilde{x}_v \\ \tilde{y}_v \\ \tilde{z}_v \end{bmatrix} = \mathbf{R}_\theta \begin{bmatrix} x_v \\ x_v \\ x_v \end{bmatrix}, \tag{6}
$$

and

$$
F_s - F_v = \hat{\mathbf{b}}_v \cdot \begin{bmatrix} \delta x \\ \delta y \\ \delta z \end{bmatrix} - \delta F + \epsilon_s, \tag{7}
$$

where $x_a, y_a, z_a$ are the three components in a geodetic reference frame of the magnetic field vector derived from of the hand-made absolute measurements of the Declination, Inclination and total field strength on the main pillar of the observatory. $\epsilon_x$, $\epsilon_y$, $\epsilon_z$ and $\epsilon_s$ are the errors that should reduce to measurement errors once the $\delta x$, $\delta y$, $\delta z$ and $\delta F$ values have been adjusted. $\epsilon_s$ includes also the errors associated with the linearization in equation (4).

The variances of the errors in the right-hand side of equations (6) and (7) are minimized iteratively by adjusting $\delta x$, $\delta y$, $\delta z$ and $\delta F$, where the iterations combined with a classic re-weighting least-squares approach allow to handle the weak non-linearity in equation (7) as well as the possible non-gaussian distributions of residuals (see e.g. Farquharson and Oldenburgh, 1998). We observe that the quality of the fit to the data depends heavily on the chosen $\theta$ angle value.

### 3.2 Application to La Réunion observatory data

The first step required to process the data is to choose the $\theta$ angle. For this, we prepared a data set that includes handmade absolute measurements $x_a, y_a, z_a$ and the total intensity measurements $F_s$ sampled every two hours, from January $1^{st}$ to June $7^{th}$, 2023. The dataset is shown in Figure 3. Absolute data can be compared with variation vector data at the same instant indicating large magnetic gradients on the observatory site, although the reference frames for the variation vector data and the

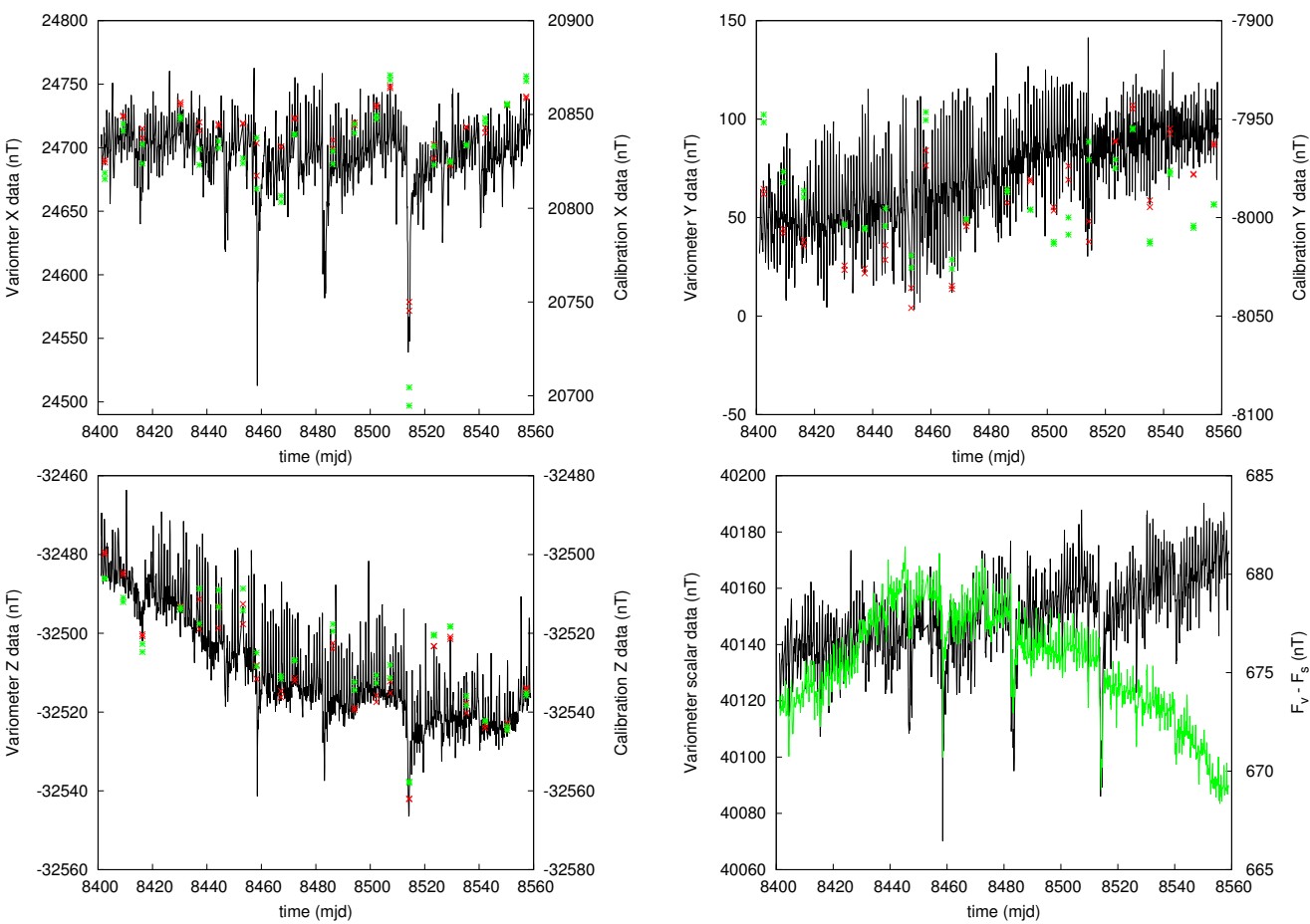

**Figure 3.** Data sets, top row $X$ and $Y$ components, bottom row $Z$ component and total intensity data. The time unit is "modified Julian day" (mjd) – i.e. decimal day number starting from January the first, 2000 at 00h00. In black are shown variation data decimated to one point every 2 hours and in red are shown variation data at the reference times corresponding to the absolute data. Scales are on the left-hand side of the figures. In green are shown the absolute data for the X,Y and Z component, and $F_v - F_s$ for the total intensity data. Scales are on the right-hand side of the figures. The $X$ and $Y$ axis of the absolute data are in geodetic reference frame, but are in the instrument reference frame for variation vector data.

absolute data are different. The magnetic field strength differences $F_v - F_s$ are of the order of $675$ nT; variations can exceed 2

nT during a single day.

Our ability to minimise the left-hand side of equations (6) and (7), by adjusting the $\delta x$, $\delta y$, $\delta z$ and $\delta F$ values, has been tested for $\theta$ values in the range $[0 : 45]$ degrees. Results are shown in Figure 4 where the misfits for the horizontal, vertical and total

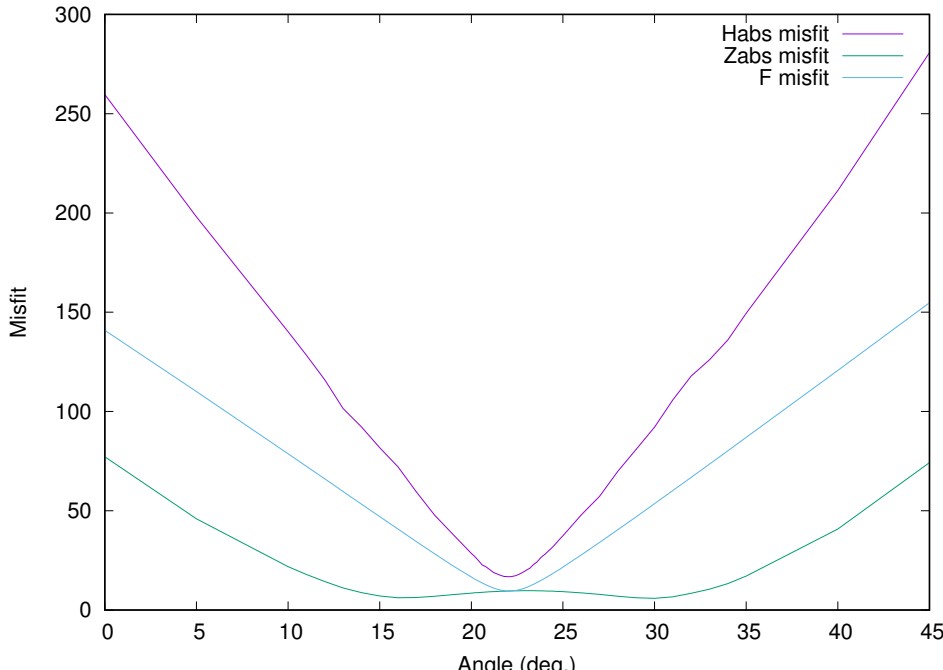

**Figure 4.** Misfits to data as a function of the $\theta$ rotation angle. The misfits for the horizontal component are shown in purple, misfits for the vertical component in green, misfits for the total intensity in blue. The misfit definitions are given in equations (8).

intensity are defined by:

$$
\begin{aligned}
M_H &= \sqrt{\sum_{\{i\}} (\epsilon_{xi}^2 + \epsilon_{yi}^2)/\sigma_i^2} \ , \\
M_Z &= \sqrt{\sum_{\{i\}} \epsilon_{zi}^2/\sigma_i^2} \ , \\
M_F &= \sqrt{\sum_{\{i\}} \epsilon_{si}^2/\sigma_i^2} \ ,
\end{aligned}
\tag{8}
$$


respectively, where the errors $\epsilon_{xi}$, $\epsilon_{yi}$, $\epsilon_{zi}$, and $\epsilon_{si}$ are defined in equations (6, 7), and $\sigma_i^2$ are the expected variances of the corresponding errors. The quantities $M_H$, $M_Z$ and $M_F$ in equations (8) are unitless and the summations are over all available absolute measurements. Figure 4 shows that the smallest misfit to the data can be achieved for $\theta = 22.0^o$. For this angle, the values of $\delta x$, $\delta y$, $\delta z$ and $\delta F$, as estimated with the approach described in section (3.1), range in the intervals $[-2402 : -2391]$

nT, $[322 : 366]$ nT, $[-21 : -8]$ nT and $[-715.5 : -713.2]$ nT, respectively. These ranges are large, particularly for $\delta y$ and $\delta z$. This occurs because we used an $L_2$ norm minimizing process that does not down-weight outliers. It is however obvious that the $\Delta F$ variations inside a day, defined by equation (4), are due to the very large $\delta x$ values.

In order to produce near-real time data with $\Delta F$ values nearly constant over a day, we decomposed the process leading to

calibrated data values for this observatory in two steps:

(i) Apply in near-real time a first correction with $\delta x_1$, $\delta y_1$, $\delta z_1$, $\delta F_1$ and $\theta_1$ values, constant over a year. These data are distributed five to ten minutes after acquisition as observatory variation data.

(ii) Apply a second correction with $\delta x_2$, $\delta y_2$, $\delta z_2$ and $\delta F_2$ varying from day-to-day, but being constant over a day. The value of $\theta_2$ remains constant over the full year and is, for 2023 and 2024, such that $\theta_1 + \theta_2 = 22^o$. These calibrated data are distributed in the following year as observatory definitive data.

The offset values $\delta x_1$, $\delta y_1$, $\delta z_1$, $\delta F_1$ can be set arbitrarily, but in order to have $\Delta F$ variations that remain small over a day, they should have values close to the $\delta x$, $\delta y$, $\delta z$, $\delta F$ intervals given above. We use:

$$
\begin{aligned}
&\delta x_1 = -2400 \text{ nT} \quad &&\delta y_1 = 280 \text{ nT} \\
&\delta z_1 = -20 \text{ nT} \quad &&\delta F_1 = 0 \text{ nT},
\end{aligned}
\tag{9}
$$

for an angle value $\theta_1 = 0.95^o$. This latter value has been set arbitrarily, but controls the values obtained for $\delta x_1$ and $\delta y_1$. To illustrate the effect of step (i) the bottom-right image of Figure 3 is shown again in Figure 5, but with the offsets applied to the variation vector data. Typical values of the magnetic field strength differences $F_v - F_s$ have changed from around 675 nT to $-713$ nT and although overall variations have similar trend and amplitude, it is clear that short-time variations have been drastically reduced within a day.

The baseline daily values: $\delta x_2$, $\delta y_2$, $\delta z_2$ and $\delta F_2$, are estimated using the same algorithm described in section (3.1), in 10 iteration, with a $\theta_2$ value set to a constant value $\theta_2 = 21.05^o$. We used a dataset consisting of 108 absolute measurements of Declination, Inclination and total Intensity, collected between 02/01/2023 and 29/01/2024, typically one double measure per week, to estimate $x_a$, $y_a$ and $z_a$ values. We assumed that these absolute data have standard variations of $0.05^o$, $0.025^o$ and 1 nT in Declination, Inclination and total Intensity, respectively. Field strength difference $F_v - F_s$ values in equation (7) are minimized at 02h00 in the morning and 22h00 in the evening, for each day of the year. Standard deviations for variation scalar and vector data were set to 400pT. The derived baseline values were assumed to be uncorrelated random variables following normal distributions. The expected mean values were set to 50 nT, 35 nT, 40 nT and $-715.5$ nT for $\delta x_2$, $\delta y_2$, $\delta z_2$ and $\delta F_2$, respectively. Variances were set to 50 nT$^2$ for $\delta x_2$, $\delta y_2$, $\delta z_2$ and 3 nT$^2$ for $\delta F_2$. Details on the algorithm and the methodology for estimating these variances are given in Lesur et al. (2017).

The estimated baseline values are shown in Figure 6 in a HDZF format (Horizontal component, Declination, Vertical down component and Total intensity). The baseline data values $H_0$, $D_0$, $Z_0$ and $F_0$, shown in red are linked to equations (6,7) by:

$$
\begin{aligned}
H_0 &= \sqrt{x_a^2 + y_a^2 - \tilde{y}_v{}^2} - \tilde{x}_v \\
D_0 &= \arctan(y_a/x_a) - \arctan(\tilde{y}_v/(H_0 + \tilde{x}_v)) \\
Z_0 &= z_a - \tilde{z}_v \\
F_0 &= F_s - F_v.
\end{aligned}
\tag{10}
$$

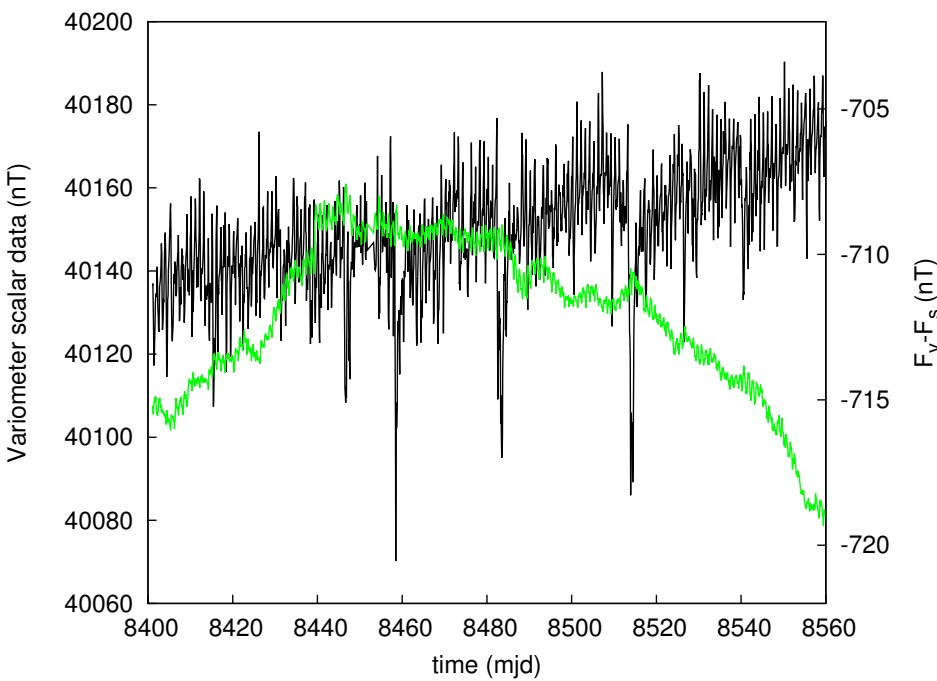

**Figure 5.** Total intensity values sampled every two hours are shown in black (left scale), and in green $F_v - F_s$ (right scale) where the rotation and offsets given in equations (9) have been applied to the raw data. The time unit is "modified Julian day" (mjd) – i.e. decimal day number starting from January the first, 2000 at 00h00.

We observed a rapid drift in the baseline values around mjd= $8680$ (i.e. October 2023), likely due to a change of environ-
ment near the variometer pillar, associated with rain or wind. Small pillar movements are also possible, as the observatory infrastructure was built only a year before, over 2022.

### 3.3 Calibrated data for year 2023

Calibrated data, estimated from the variation vector data using the baseline values of Figure 6, are presented in Figure 7. The horizontal component presents the expected large amplitude fast variations that are associated with perturbations of the
ionosphere-magnetosphere system by the sun activity. There is nonetheless a small trend of increasing intensity by roughly 50nT over 2023. The trend is even stronger on the vertical component that increases by around $100$ nT in absolute value in the year. The combined effect of these variations produces an increase of the magnetic field strength approximately from 39420 nT to 39530nT – i.e. an annual variation of the order of $100$ nT/y that corresponds roughly to what was predicted by the International Geomagnetic Reference Field (IGRF) version 13 (Alken et al., 2021): $104$ nT/y in 2023. For this same IGRF, and same
location as the observatory, the expected variation in the vertical component is largely underestimated and so is the expected variation in Declination. The values of the field components as provided by the IGRF model, differ also significantly from the

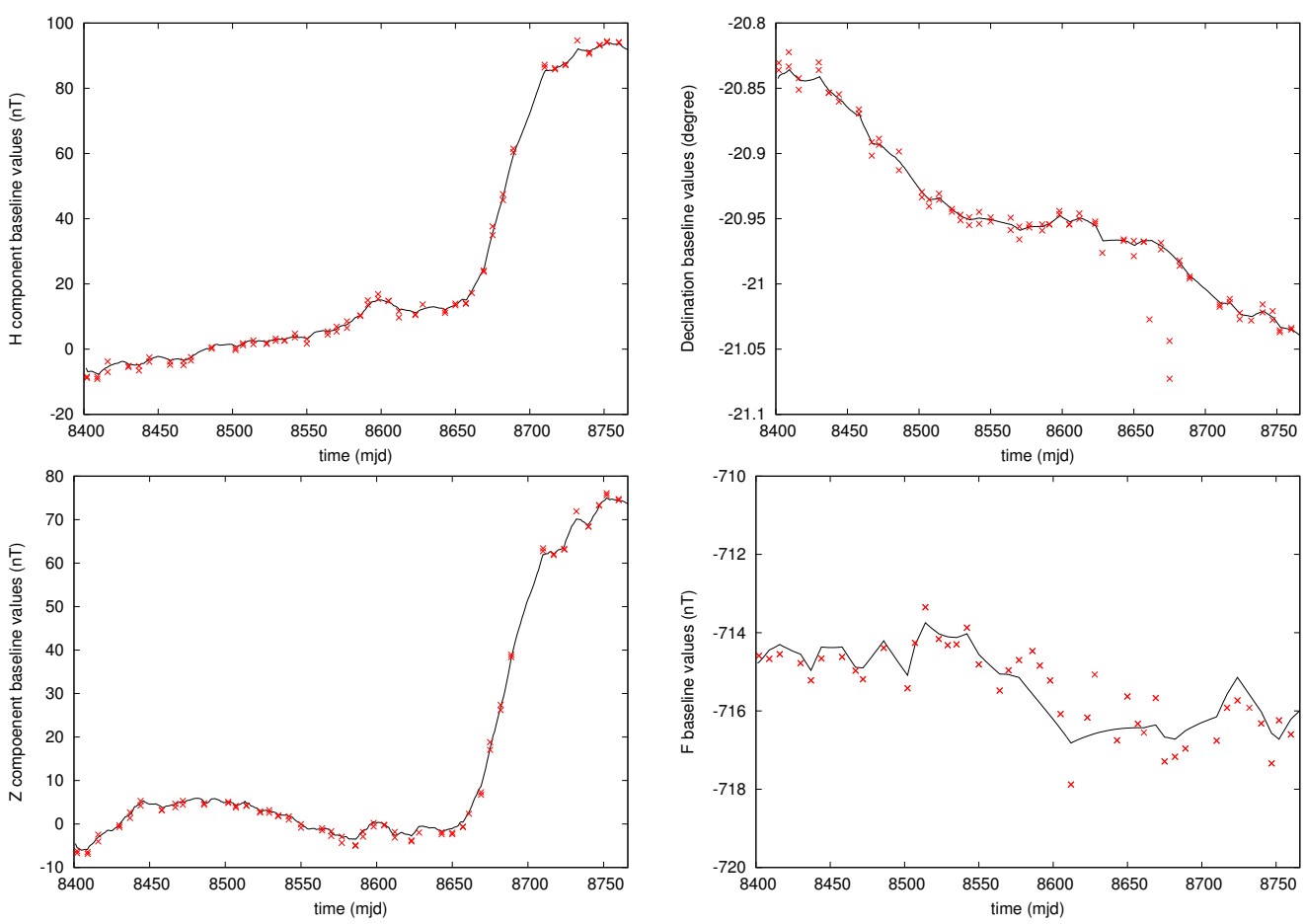

**Figure 6.** Estimated baseline values of La Réunion observatory, presented in a HDZF format, for year 2023. The time unit is "modified Julian day" (mjd) – i.e. decimal day number starting from January the first, 2000 at 00h00. In red are shown the values derived from absolute observations and in black the estimated daily values.

measured values. The IGRF gives on 2023-06-01: $-19.515^o$ in Declination, 22690 nT in the Horizontal component, $-31458$ nT in the Vertical component and 38787 nT in Total intensity. The observed differences are easily explained by the strong lithospheric signal generated by the surrounding volcanic rocks that is not accounted for in the IGRF.


On the same Figure 7 the bottom right plot presents also, in green, the calculated $\Delta F$ values for the year. The scale is given on the right side of the plot. Values vary well inside 1 nT around zero. This is a clear indication that the process applied to estimate the baseline values is a success.

As a final test to assess the quality of the observatory recorded signal, we computed the Discrete Fourier Transform (DFT) amplitude spectra of the vertical and horizontal components of the observed, de-trended, magnetic field variation second-data over six months from July to December 2023 (see Figure 8). Over this time interval, only five consecutive second-data records were missing. The gap has been filled by linear interpolation. Of course, as only variation data have been used, the time series have not been treated to remove possible anthropological noise, and furthermore the longest periods of the spectra are not

reliable as the baseline correction has not been applied. Strong peaks are obvious at periods of 24h, 12h, 8h, 6h, 4.8h and 4h (see also Figure 9 for a zoom on periods range from 4 hours to 13 hours). These periods correspond to the S1 to S6 Solar diurnal tidal constituents (see e.g. Love and Rigler (2014) regarding tidal signals in observatory data). However, in magnetic data these peaks result mainly from the rotation of the Earth inside the magnetosphere combined with the signal associated with the $S_q$ current system in the Ionosphere. There are no other clear tidal periods peaking out of the spectra outside, possibly,

at 12.42 hours for the M2 (Lunar semi-diurnal) tide in Figure 9 (left). However, this period does not correspond to a peak in the horizontal component spectra. There are few peaks in the lowest periods of Figures 8, in particular for a period of 5 seconds. This is clearly due to a cross talk of the scalar/vector electronics as 5 seconds is our sampling period for scalar variometer data. At even shorter periods the spectra collapse due to the filtering of the lowest periods applied to second data, as recommended in INTERMAGNET (2020). Overall this Fourier analysis does not reveal major difficulties in the observatory data. There is a

relatively low level of anthropological noise at the La Réunion observatory site.

## 4    Conclusions

We presented the setting and location of the La Réunion Island observatory, and the data processing algorithms applied to compensate for the effects of the large magnetic gradients typical for volcanic islands. This observatory has been set to fill a geographical gap in the Indian Ocean part of the global observatory network. Calibrated data have been estimated for the full

year 2023.

Similar to other observatories situated on a volcanic island, the magnetic field induced by local geology is significant and exhibits steep gradients. We have shown that one of the effects of these gradients is a variation during a single day of the differences of field strength between two sites only few meters apart. To reconcile the magnetic field strength observed by the

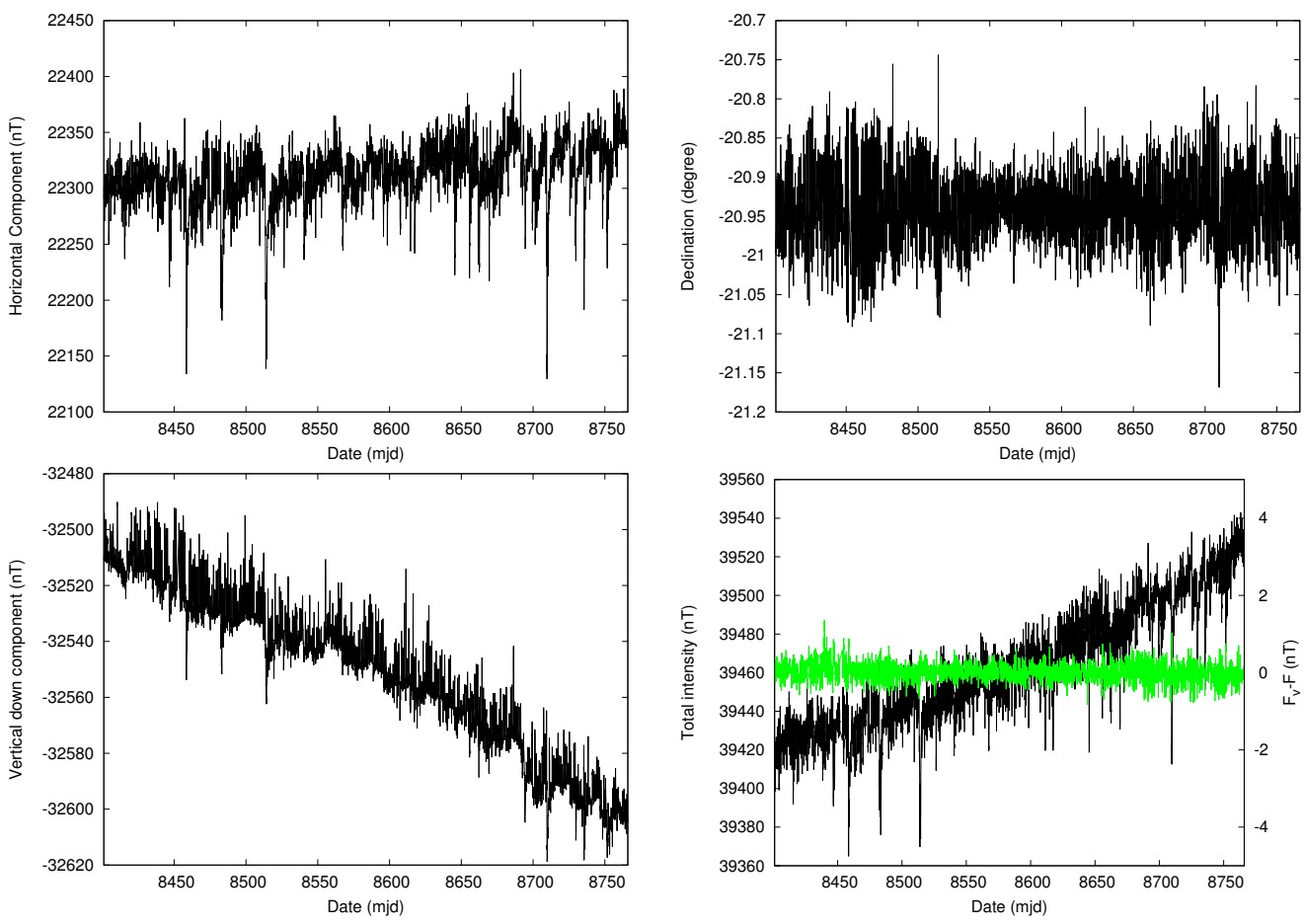

**Figure 7.** Estimated calibrated hourly mean magnetic data at La Réunion observatory. The date unit is "modified Julian day" (mjd) – i.e. decimal day number starting from January the first, 2000 at 00h00. The data are presented in a HDZF format, for the whole year 2023. In green are shown the $\Delta F$ values derived from the hourly means of the vector and scalar values. The corresponding scale is shown on the right side of the plot.

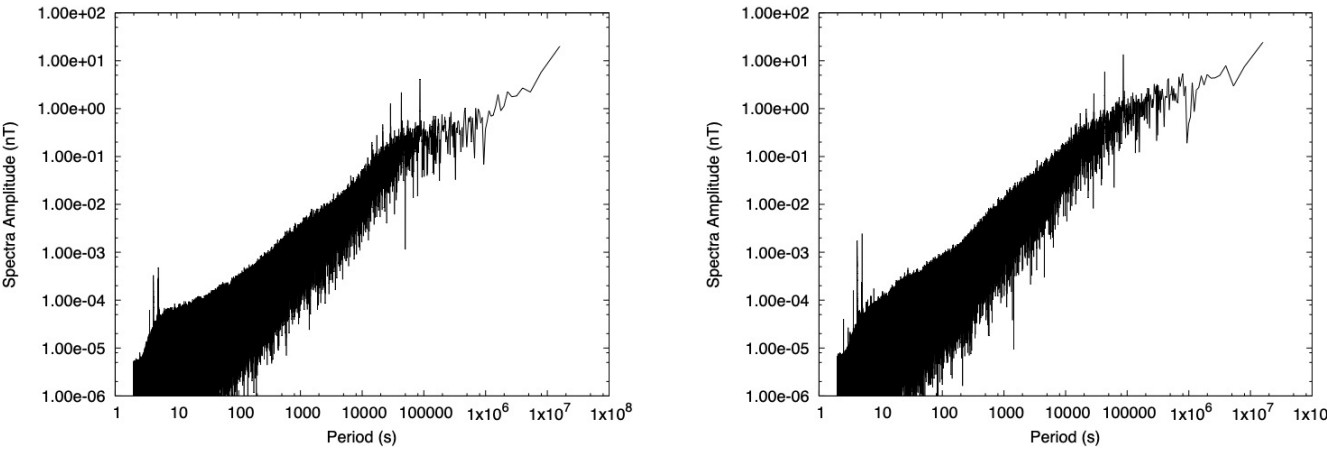

**Figure 8.** DFT amplitude spectra derived from time series of second-data recorded in La Réunion magnetic observatory, from $1^{st}$ of July 2023 to the $31^{st}$ of December 2023. Left, spectrum derived from the vertical down component, Right, spectrum derived from the horizontal component.

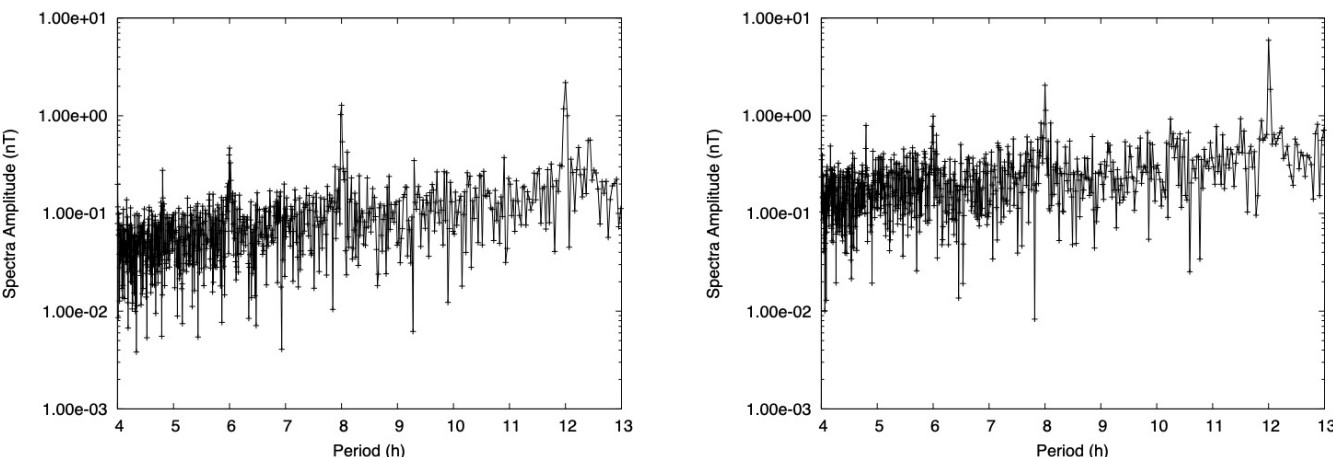

**Figure 9.** Zoom on the 4–13 hour period range of the DFT amplitude spectra derived from second-resolution time series, recorded in la Réunion magnetic observatory, from $1^{st}$ of July 2023 to the $31^{st}$ of December 2023. Left, spectrum derived from the vertical down component, Right, spectrum derived from the horizontal component.

variation vector magnetometer with the magnetic field strength measured by the variation scalar magnetometer, one has simply to estimate the large local contribution of the magnetized rocks to the observed vector magnetic field. These estimates do not have to be very accurate and in the case of La Réunion observatory they are of -2400 nT, 280 nT and -20 nT in the $X$, $Y$ and $Z$ vector component, respectively, in the sensor reference frame.

The obtained calibration parameters – i.e. the "baseline values", show a strong drift within the year 2023, particularly during the month of October. This is very likely due to the fact that pillars have been built only recently on the observatory site and are still settling. We therefore expect the baseline values to stabilise and present only minor drifts in the years to come. For 2023, we are however confident that the calibrated observatory vector data are reasonably accurate as the drift of the $\delta F$ values remains weak, and the fit to the absolute observation is good.

The IAGA code given to this observatory is : REU.

*Data availability.* Definitive and variation data derived from La Réunion (REU) observatory are available on www.bcmt.fr

## Appendix A: Derivation of equation (4)

The total intensity on the variometer pillar is:

$$F_v = \sqrt{x_v^2 + y_v^2 + z_v^2}, \tag{A1}$$

whereas on the observatory main pillar it is:

$$F_p = \sqrt{(x_v + \delta x)^2 + (y_v + \delta y)^2 + (z_v + \delta z)^2}. \tag{A2}$$

The latter quantity can be approximated by:

$$F_p \simeq \sqrt{x_v^2 + y_v^2 + z_v^2 + 2(x_v \delta x + y_v \delta y + z_v \delta z)}, \tag{A3}$$

where terms in second order of $\delta x$, $\delta y$ or $\delta z$ are neglected. It follows that:

$$F_p \simeq F_v \sqrt{1 + 2(x_v \delta x + y_v \delta y + z_v \delta z)/(x_v^2 + y_v^2 + z_v^2)}, \tag{A4}$$

or alternatively:

$$F_p \simeq F_v [1 + (x_v \delta x + y_v \delta y + z_v \delta z)/F_v^2], \tag{A5}$$

where the quantity $(x_v \delta x + y_v \delta y + z_v \delta z)/F_v^2$ is assumed to be small. Here again, higher order terms are neglected. The difference $F_p - F_v$ is therefore:

$$F_p - F_v \simeq (x_v \delta x + y_v \delta y + z_v \delta z)/F_v, \tag{A6}$$

and noticing that the right-hand side of equation (A6) is the result of a vector scalar product, equation (4) follows. The same result can be obtained by simply using a first order Taylor series for the total field intensity at the main observatory pillar:

$$F_p \simeq F_v + \frac{\partial F_v}{\partial x_v}\delta x + \frac{\partial F_v}{\partial y_v}\delta y + \frac{\partial F_v}{\partial z_v}\delta z. \tag{A7}$$

325 *Author contributions.* B.H. and V.L. wrote this manuscript, the observatory has been technically realised by B.H., A.T. and F.P., the data processing has been defined by V.L. and achieved by B.H. and V.L. All authors contributed to the project organization and realisation.

*Competing interests.* The authors do not have competing interests

*Acknowledgements.* The authors would like to thank the ONF (Office National des Forêts) and the OVPF (Observatoire Volcanologique du Piton de la Fournaise) for their collaboration in this project.

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
