# Peer review of "A new magnetic observatory in La Réunion Island – meeting data quality requirements in a volcanic island setting"

_EGUsphere, 2025_

## Referee Comment (RC1)

**Review by Tanja Petersen**

The manuscript "A New Magnetic Observatory in La Réunion Island" discusses the establishment of a new observatory on a volcanic island and how challenges posed by the volcanic setting are addressed through robust data processing and quality control measures.

Observatories on remote islands are essential for filling gaps in the global observatory network, particularly in oceanic regions. Volcanic rocks generate strong local magnetic field gradients, leading to non-uniform magnetic fields. Since the proton and fluxgate magnetometers are not co-located, they measure different fields, causing daily variations in the F difference (the difference between the total field calculated from the fluxgate magnetometer components and the total field measured by the proton magnetometer). To mitigate this, the authors apply a constant offset to each vector component, effectively correcting the measurements. The manuscript demonstrates that this correction ensures the observatory data meet quality standards without significant noise contamination. The authors successfully estimate baseline values and their thorough processing approach significantly improves the recorded signal quality.

This work represents a valuable contribution to the special issue on "Geomagnetic Observatories, Their Data, and the Application of Their Data." The scientific approach is sound, the methods are well-established, and the results and conclusions are well-presented.

However, I do have a few minor scientific questions and specific comments I would like to make:

Line 88-90 – Please make very clear why you needed to make this compromise. Explain that the fluxgate instrument is very sensitive to possible pillar movement; having the pillar move by the tiniest amount would result in measuring a different magnetic field, especially in an environment with large gradients. Therefore, you want the pillar to be stable – at least long term, once the ground around the pillar has settled down and the position has stabilized. A short pillar means higher stability but also being closer to the magnets (the volcanic rocks). In my view it is a valuable point for the reader to understand.

Line 94 – What is the horizontal distance between the scalar and the fluxgate instrument?

Line 69 – You mention that your first step was "surveys of potential sites". I think explaining more here would be of value and interest. Did you survey the vertical gradients in the area? Did you run a grid survey or performed spot measurements at potentially suitable locations? Using the survey results to establish a gradient map of the area (e.g. a map displaying vertical gradients (nT/m) in contours) or a map of total field measurements taken would show how non-homogeneous the magnetic field is in the volcanic setting. It would visualize the challenges you are facing by establishing an observatory on a volcanic island.

Line 99 – Some observatories do use an additional scalar magnetometer for the handmade absolute measurements. As you did not allocate a section to it I assume you are using the variation scalar magnetometer data for your absolute measurements of the magnetic field strength. However, adding a sentence to clarify this would be good.

Line 180 - please explain the right side (with:..) of the equation. Is that your assumption? Not clear to me.

Line 185 – please explain what you mean by observation errors.

Although I do like your short title, you could add "A new magnetic observatory in La Réunion Island – meeting data quality requirements in a volcanic island setting" (or something similar) to include the data processing content of your paper. It would be nice to have your title reflect the challenges and your thorough approach with regards to meeting Intermagnet observatory standards.

Overall, please make sure you keep a clear structure and be consistent with your naming of different types of instrumentation and measurements. This will make it easier for a reader who is less familiar with details of observatory observations to follow and understand. I have added a few suggestions / attempts to my list of suggestions below.

In addition, here are some suggestions for specific technical corrections or improvements the authors could make:

**Abstract.**

Line 2 – define OVPF: Observatoire Volcanologique du Piton de la Fournaise (OVPF)

Line 3 – This magnetic observatory.... monitoring the evolution of the Earth's magnetic field in that region.

Line 9 & 11 – This sentence could mean two different things: please clarify if you're adding -2400 nT, 280 nT and -20 nT to each component (X, Y & Z) but in a systematic way across each day, OR if you're adding, e.g. -2400 nT to X, 280 nT to Y in a systematic way. The word "systematic" could be misleading when only reading the abstract. Maybe write: ..., the constant offset values of -2400 nT, 280 nT and -20 nT have been systematically determined and are being added to the X, Y and Z magnetic field components, respectively, prior to the data distribution.

**Introduction.**

Line 29-38 - Moving this paragraph to being placed after the one where you describe the Indian Ocean setting and why La Reunion Island provides a good location would keep the paragraphs that are about the 'why it is good to have an observatory here' together.

Line 37 – ...located on the observatory main pillar, the location all observatory data is in reference to. --- or something along the line.

Line 58 - .... for processing magnetic survey or variometer station data – removing the plural from surveys makes data refer to both data sources.

Line 60 - ....are described in detail --– you're already describing location and setting above, so you want to distinguish from that.

La Réunion observatory setting.

Line 65 - La Réunion Island is a volcanic island...

Line 67 - ...the island, between two volcanoes.

Line 67 – Aeromagnetic surveys that were flown show...

Line 68 - ...in three steps:...

Line 71-73 – The land, covered by forest, is owned....The area has little elevation change, the forest is ....

Line 74 – One visual target ... --- the "without magnetic constraints" needs to either be explained more or differently or taken out (and addressed later if needed).

Line 75 - ... 40 m from the observatory main pillar.

Line 76-77 - A good grounding to avoid lightning strikes and a strong attachment of the built infrastructure to the ground to provide good resistance to hurricanes has been necessary.

Line 86 - ...pillar; because of...

Line 93-94 - These types of instruments are sensitive to magnetic field gradients, but less sensitive to possible pillar movement and therefore it has been set at 1.7m above ground...

Line 98 - Each absolute observation is a combination of a series ...

Line 99 – The angle measurements...

Line 103 – ....for handmade absolute measurements... -- try to stay consistent with your naming.

Line 104 - The variation scalar magnetometer....

Line 107 - ...by an IPGP in-house built 'ENO4' data logger... -- unless I am wrong and it is a conventional data logger that I have never heard about.

Line 108 - ...via GSM digital cellular signal... --- for the reader to get pointed into the right direction to what GSM means. Or you could write it out.

Figure 2 – In front, the observatory main pillar during a absolute measurement training session is shown. --- consistency in naming again. For the rest of the caption, please try to tidy up the sentence. Or, what you could do is, add numbers in white to the photo – above each feature that you want to name. That would help the reader find, e.g. the target pole, and help you to tidy up the description. You could also make use of parentheses, e.g. .... the grey vertical PVC tube (partly hidden behind the trees) containing .... And: ... variation vector magnetometer (covered with a thermal white blanket).

Line 113 – Only the GSM ...

**Data Processing.**

Line 120 – observatory main pillar (or main pillar) --- you could define "main pillar" here so that you do can use it instead of "observatory main pillar" throughout the rest of the processing section.

Line 142 - ...Sq current system...

Line 149 – due to, e.g., temperature....

Line 150 - ...at La Réunion observatory.

Line 153 – briefly recall...

Line 155 – vector field $b_p$ estimated...

Line 158 - ..., namely θ,...

Line 162 - ...that θ is close to ...

Line 168 – ...$F_s$ taken from ...  ---- its not really estimated. Or is it? Instead you simply use a variation scalar datum?

Line 170-171 – ...to be constant would require the magnetic field gradients to be small ... --- or something similar? I stumbled across your sentence.

Line 173-175 – This approximation was validated by placing an additional scalar magnetometer on the main pillar for one day (23$^{rd}$ June, 2023) to record the magnetic field strength simultaneously on the two locations, main pillar and variation scalar magnetometer pillar. The difference between the measurements did not exceed 0.5 nT.

Line 184 - ...on the main pillar...

Line 191 - ...the quality of the fit...

Line 193 - ...is to choose the θ angle value. For this we compose a dataset that...

Line 194 - ...January 1$^{st}$ to June 7$^{th}$, 2023.

Line 195 - ...dataset...   --- also: leave the colour description to the figure caption.

Line 198 - ...675 nT; variations can exceed 2 nT ...

Line 212 - ...produce near-real time...

Line 223 - ...effect of step...

Line 265 - ... only five consecutive...

Line 273 - ...Figure 9 (left).

**Conclusion.**

Line 280 – We presented the setting and location of the La Réunion Island observatory, and the data processing algorithms applied to compensate for the affects of the large magnetic gradients typical for volcanic islands.

Line 281 - ...gap in the Indian Ocean part of the global observatory network.

Line 284 – Similar to other observatories....

Line 286-287 - ...observed by the variation vector magnetometer...

Line 288 - ...contribution the magnetized rocks have to the vector magnetic field in order to be effective.

Line 290 - ...component, respectively, ...

Line 293 - ...observatory site and are still settling.

---

## Author Response (AR1)

Dear Editor,

We would like to thank the two reviewers for their helpful comments regarding the manuscript. The reviewers pointed to few minor issues that we have all addressed. They lead to some changes in the manuscript that are highlighted in red in the associated file. Below you will find further few comments regarding each of these points. The original reviewer comments are in bold fonts.

We point that as suggested by the reviewer 1, we have modified the title of the manuscript.

**Reviewer-I**

**Line 88-90 – Please make very clear why you needed to make this compromise. Explain that the fluxgate instrument is very sensitive to possible pillar movement; having the pillar move by the tiniest amount would result in measuring a different magnetic field, especially in an environment with large gradients. Therefore, you want the pillar to be stable – at least long term, once the ground around the pillar has settled down and the position has stabilized. A short pillar means higher stability but also being closer to the magnets (the volcanic rocks). In my view it is a valuable point for the reader to understand.**

In this part where we describe the data types, it is better not to discuss in details this compromise between pillar stability and strong magnetic rocks contribution. We therefore move this part of the text further down in the next paragraph and pointed to the risk linked to hurricane and the necessity of robust installation.

**Line 94 – What is the horizontal distance between the scalar and the fluxgate instrument?**

Around 6 meters. The distances are now reported on the observatory plan that has been added to the paper.

**Line 69 – You mention that your first step was "surveys of potential sites". I think explaining more here would be of value and interest. Did you survey the vertical gradients in the area? Did you run a grid survey or performed spot measurements at potentially suitable locations? Using the survey results to establish a gradient map of the area (e.g. a map displaying vertical gradients (nT/m) in contours) or a map of total field measurements taken would show how non- homogeneous the magnetic field is in the volcanic setting. It would visualize the challenges you are facing by establishing an observatory on a volcanic island.**

We have modified the text while trying not to put too much emphasis on this point that remains a minor point for this paper that is more focused on the data processing and presenting the observatory.

**Line 99 – Some observatories do use an additional scalar magnetometer for the handmade absolute measurements. As you did not allocate a section to it I assume you are using the variation scalar magnetometer data for your absolute measurements of the magnetic field strength. However, adding a sentence to clarify this would be good.**

No. Scalar absolute measurements are made on the observatory main pillar, before and after, hand-made declination and inclination measurements. The type of instrument used has been added to the text.

**Line 180 - please explain the right side (with:..) of the equation. Is that your assumption? Not clear to me.**

This righthand side equation simply gives the definition of the \tilde quantities. We replaced "with" with "where" in the new version.

These first reviewer comments are followed by a series of minor remarks that we almost all implement in the document.

**Reviewer-II:**

**Line 9 -- "values vary during a day" How much?**

More than ~2nT. This is now indicated in the text.

**Line 13 -- "although the baseline values present strong variations" How much is the variations observed in the baseline?**

70 nT, this is now given in the abstract

**Line 17-18 – "There are currently around 120 magnetic observatories around the world collecting data, most of them being part of INTERMAGNET (Love and Chulliat, 2013);" Please update the statement with the latest information you can refer latest paper.**

We are not aware of recent published paper giving a review of the INTERMAGNET network. We will be happy to include the reference if such a paper exists.

**Line 37 -- What do you mean of calibration vector data? It is different from definitive data?**

We have not been precise enough regarding what is described as "calibration", "absolute" and "definitive". We modified the text to systematically use "absolute" data (or measurements) in place of "calibration". We prefer using "calibrated data" to "definitive data" because calibrating data does not imply that they should not change in the future. Deciding if calibrated data are "definitive" is under the responsibility of the scientists or technicians in charge of the observatories. The text has been modified.

**Line 67-68 -- Please provide the details of the magnetic gradient values at selected area for observatory.**

This point has been raised by the reviewer-1. Our answer is the same as for reviewer-1.

**Line 80 -- The figure 2 does contain much information. Please include a layout map of observatory site with details of locations, instruments and their distances between them and details of the construction.**

A schematic map of the observatory has been added.

**Line 95 -- Please remove "These types of data are called the variation scalar data." I don't feel that it is necessary.**

There is often a confusion between the different types of data that are produced in observatories, in particular from scientists that are not used to observatory operations. We here define and name each kind of data, and use these "definitions" throughout the manuscript. We would prefer keeping these "definitions" for the manuscript clarity. The sentence has not been erased.

**Line 99-100 -- Please remove "The data angles are completed with absolute measurements of the magnetic field strength." I felt it does not contain necessary information.**

Often in observatories, the magnetic field strength is not measured before and after declination and inclination measurements. These measurements are made in La Réunion observatory and we think it is worth pointing to this fact. We have indicated the instrument type at the request of the reviewer 1.

**Line 102 -- Rewrite it "As partly described above, to set the instruments two pillars in fibered concrete were build"**

The sentence has been reorganized

**Line 102-103 -- "at chest height," write measurements standard way.**

This is roughly 1.4m. The text has been modified.

**Line 103-104 -- "a small one to rest the variation vector magnetometer in a box on the ground," rewrite please. Here it is not clearly written about the sensor hut/vault/room. Please write it clearly.**

There is no hut or vault to set the vector magnetometer but a simple box, with a proper thermic insulation. The text has been modified to clarify this point.

**Line 125 –"strong magnetic gradient" what are probable gradients observed in this site during quiet and disturbed days.**

We don't think we need to indicate probable gradient values at this point in the paper. These gradients are estimated during the processing. As far as we can observe, there is no major differences between quiet or disturbed days. The values of the gradients are directly linked to the observed site differences that are, as indicated in the paper, of the order of 2500nT between the main pillar and vector variometer pillar and 718 nT between the main pillar and the scalar variometer pillar.

**Line 129 – " magnetometer reference frame" explain.**

This is the frame defined by the three components of the vector (fluxgate) magnetometer. The text has been modified to clearly define this reference frame.

**Line 157 – "what is rotation matrix for an angle and how do you define it here" Explain clearly.**

Of course, we could add -- e.g. in appendix, the definition of a rotation matrix, but this is expected to be known by scientist reading this kind of paper. The definition of a rotation matrix is given in most academic books of introduction to mathematics. It can also be easily accessed through internet. We will follow here the decision of the editor regarding this point.

**Line 173–175 – what is the periodicity of offset measurements " This approximation can be however tested using an additional scalar magnetometer by recording the magnetic field strength simultaneously on the observatory main pillar and with the variation scalar magnetometer." And explain the equation used clearly.**

These measurements were done only once at 0.2 Hz on the 23rd of June. We modified the text and also slightly the equations as required by reviewer-1. We hope that it is now clearer.

**Line 197 – Figure 3, It is not clear what is x-axis means. Please do figure description and explain in detail. I suggest to include for all figures.**

The time unit is "modified Julian Day", commonly used in satellite magnetic data. The time is counted in decimal days starting from the 01/01/2000 at 00h00. We modified the Figure 3,5,7 and 7 captions to include this information.

**Line 202 – Figure 4, please explain the about figure, what is the right side axis? How did you calculated the values mentioned at the line 207. Please provide a clear description of the formula at line 203.**

There is no quantity described by the figure-4 right side axis. The left side axis are the misfit values calculated using equations (8) as explained in the caption. We modified the text to point to the algorithm described in section 3.1, for the estimation of the baseline values.

**Line 233 – Please correct them "02h00 in the morning and 22h00" Line 235 – Please correct "... 50 nT$_2$ for ... 3 nT$_2$**

We do not understand what the reviewer wants us to modify. We have not changed the text.

**Line 243 – May I know the how the pillar had constructed and can you explain the probability of the pillar movement. How do you check the same and do you have any example?**

The constructed pillar is made of a large concrete block underground, and it is not rare that for the first few years, the pillar moves slightly before it stabilize. We already observed this effect on newly built pillars, in 2016, in Chambon-la-forêt observatory. We did not modify the text.

---

## Author Response (AR2)

**Response to the editor comments:**

Editor comments are in Italic fonts.

*The comment "Line 185 - please explain..." by Reviewer-1 has not been addressed yet. Please consider it and give some response.*

The errors are those of equations 6 and 7. For arbitrary set values of $\delta x$, $\delta y$, $\delta z$ and $\delta F$, the errors include the actual measurement errors and those linked to possible erroneous values for $\delta x$, $\delta y$, $\delta z$ and $\delta F$. After fitting the data, in principle only measurement errors remain. We used the term "observation errors" that we have changed to "measurement errors" in the revised manuscript.